# Learning when to communicate at scale in Multiagent Cooperative and Competitive Tasks

**Amanpreet Singh**[*]
New York University
Facebook AI Research[†]
amanpreet@nyu.edu

**Tushar Jain**[*]
New York University
tushar@nyu.edu

**Sainbayar Sukhbaatar**
New York University
Facebook AI Research[†]
sainbar@cs.nyu.edu

## Abstract

Learning when to communicate and doing that effectively is essential in multi-agent tasks. Recent works show that continuous communication allows efficient training with back-propagation in multiagent scenarios, but have been restricted to fully-cooperative tasks. In this paper, we present Individualized Controlled Continuous Communication Model (IC3Net) which has better training efficiency than simple continuous communication model, and can be applied to semi-cooperative and competitive settings along with the cooperative settings. IC3Net controls continuous communication with a gating mechanism and uses individualized rewards for each agent to gain better performance and scalability while fixing credit assignment issues. Using variety of tasks including StarCraft BroodWars[TM] explore and combat scenarios, we show that our network yields improved performance and convergence rates than the baselines as the scale increases. Our results convey that IC3Net agents learn when to communicate based on the scenario and profitability.

## 1 Introduction

Communication is an essential element of intelligence as it helps in learning from others experience, work better in teams and pass down knowledge. In multi-agent settings, communication allows agents to cooperate towards common goals. Particularly in partially observable environments, when the agents are observing different parts of the environment, they can share information and learnings from their observation through communication.

Recently, there have been a lot of success in the field of reinforcement learning (RL) in playing Atari Games (Mnih et al., 2015) to playing Go (Silver et al., 2016), most of which have been limited to the single agent domain. However, the number of systems and applications having multi-agents have been growing (Lazaridou et al., 2017; Mordatch & Abbeel, 2018); where size can be from a team of robots working in manufacturing plants to a network of self-driving cars. Thus, it is crucial to successfully *scale* RL to multi-agent environments in order to build intelligent systems capable of higher productivity. Furthermore, scenarios other than cooperative, namely semi-cooperative (or mixed) and competitive scenarios have not even been studied as extensively for multi-agent systems.

The mixed scenarios can be compared to most of the real life scenarios as humans are cooperative but not fully-cooperative in nature. Humans work towards their individual goals while cooperating with each other. In competitive scenarios, agents are essentially competing with each other for better rewards. In real life, humans always have an option to communicate but can choose when to actually communicate. For example, in a sports match two teams which can communicate, can choose to not communicate at all (to prevent sharing strategies) or use dishonest signaling (to misdirect opponents) (Lehman et al., 2018) in order to optimize their own reward and handicap opponents; making it important to learn when to communicate.

---

[*]Equal contribution.
[†]Current affiliation. This work was completed when authors were at New York University.

Teaching agents how to communicate makes it is unnecessary to hand code the communication protocol with expert knowledge (Sukhbaatar et al., 2016)(Kottur et al., 2017). While the content of communication is important, it is also important to know when to communicate either to increase scalability and performance or to increase competitive edge. For example, a prey needs to learn when to communicate to avoid communicating its location with predators.

Sukhbaatar et al. (2016) showed that agents communicating through a continuous vector are easier to train and have a higher information throughput than communication based on discrete symbols. Their continuous communication is differentiable, so it can be trained efficiently with back-propagation. However, their model assumes full-cooperation between agents and uses average global rewards. This restricts the model from being used in mixed or competitive scenarios as full-cooperation involves sharing hidden states to everyone; exposing everything and leading to poor performance by all agents as shown by our results. Furthermore, the average global reward for all agents makes the credit assignment problem even harder and difficult to scale as agents don't know their individual contributions in mixed or competitive scenarios where they want themselves to succeed before others.

To solve above mentioned issues, we make the following contributions:

1. We propose **Individualized Controlled Continuous Communication Model** (IC3Net), in which each agent is trained with its individualized reward and can be applied to ***any scenario*** whether cooperative or not.
2. We empirically show that based on the given scenario–using the gating mechanism–our model can learn **when to communicate**. The gating mechanism allows agents to block their communication; which is useful in competitive scenarios.
3. We conduct experiments on different scales in three chosen environments including StarCraft and show that IC3Net outperforms the baselines with performance gaps that **increase** with scale. The results show that *individual rewards* converge **faster** and better than *global rewards*.

## 2 RELATED WORK

The simplest approach in multi-agent reinforcement learning (MARL) settings is to use an independent controller for each agent. This was attempted with Q-learning in Tan (1993). However, in practice it performs poorly (Matignon et al., 2012), which we also show in comparison with our model. The major issue with this approach is that due to multiple agents, the stationarity of the environment is lost and naïve application of experience replay doesn't work well.

The nature of interaction between agents can either be cooperative, competitive, or a mix of both. Most algorithms are designed only for a particular nature of interaction, mainly cooperative settings (Omidshafiei et al., 2017; Lauer & Riedmiller, 2000; Matignon et al., 2007), with strategies which indirectly arrive at cooperation via sharing policy parameters (Gupta et al., 2017). These algorithms are generally not applicable in competitive or mixed settings. See Busoniu et al. (2008) for survey of MARL in general and Panait & Luke (2005) for survey of cooperative multi-agent learning.

Our work can be considered as an all-scenario extension of Sukhbaatar et al. (2016)'s CommNet for collaboration among multiple agents using continuous communication; usable only in cooperative settings as stated in their work and shown by our experiments. Due to continuous communication, the controller can be learned via backpropagation. However, this model is restricted to fully cooperative tasks as hidden states are fully communicated to others which exposes everything about agent. On the other hand, due to global reward for all agents, CommNet also suffers from credit assignment issue.

The Multi-Agent Deep Deterministic Policy Gradient (MADDPG) model presented by Lowe et al. (2017) also tries to achieve similar goals. However, they differ in the way of providing the coordination signal. In their case, there is no direct communication among agents (actors with different policy per agent), instead a different centralized critic per agent – which can access the actions of all the agents – provides the signal. Concurrently, a similar model using centralized critic and decentralized actors with additional counterfactual reward, COMA by Foerster et al. (2018) was proposed to tackle the challenge of multiagent credit assignment by letting agents know their individual contributions.

Vertex Attention Interaction Networks (VAIN) (Hoshen, 2017) also models multi-agent communication through the use of Interaction Networks (Battaglia et al., 2016) with attention mechanism (Bahdanau et al., 2015) for predictive modelling using supervised settings. The work by Foerster

et al. (2016b) also learns a communication protocol where agents communicate in a discrete manner through their actions. This contrasts with our model where multiple continuous communication cycles can be used at each time step to decide the actions of all agents. Furthermore, our approach is amenable to dynamic number of agents. Peng et al. (2017) also attempts to solve micromanagement tasks in StarCraft using communication. However, they have non-symmetric addition of agents in communication channel and are restricted to only cooperative scenarios.

In contrast, a lot of work has focused on understanding agents' communication content; mostly in discrete settings with two agents (Wang et al., 2016; Havrylov & Titov, 2017; Kottur et al., 2017; Lazaridou et al., 2017; Lee et al., 2018). Lazaridou et al. (2017) showed that given two neural network agents and a referential game, the agents learn to coordinate. Havrylov & Titov (2017) extended this by grounding communication protocol to a symbols's sequence while Kottur et al. (2017) showed that this language can be made more human-like by placing certain restrictions. Lee et al. (2018) demonstrated that agents speaking different languages can learn to translate in referential games.

## 3 MODEL

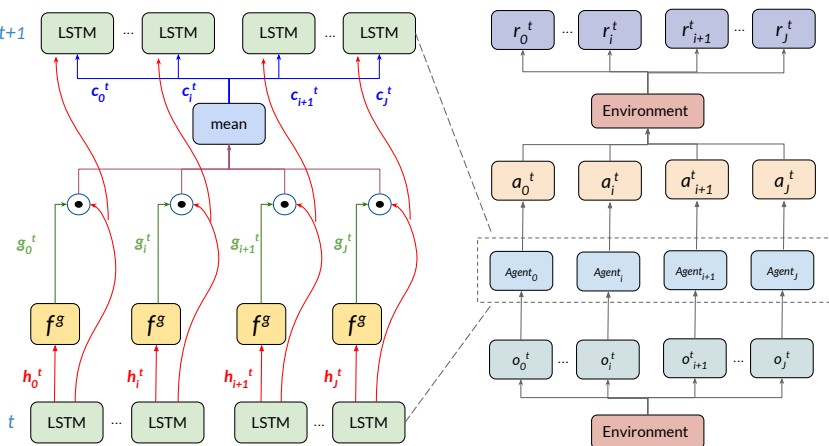

Figure 1: **An overview of IC3Net**. **(Left)** In-depth view of a single communication step. LSTM gets hidden state, $h_t$ and cell state, $s_t$ (not shown) from previous time-step. Hidden state $h_t$ is passed to Communication-Action module $f_g$ for a communication binary action $g_t$. Finally, communication vector $c_t$ is calculated by averaging hidden states of other active agents gated by their communication action $ac_t$ and is passed through a linear transformation C before fed to LSTM along with the observation. **(Right)** High-level view of IC3Net which optimizes individual rewards $r^t$ for each agent based on observation $o^t$.

In this section, we introduce our model **Individualized Controlled Continuous Communication Model (IC3Net)** as shown in Figure 1 to work in multi-agent cooperative, competitive and mixed settings where agents learn what to communicate as well as when to communicate.

First, let us describe an independent controller model where each agent is controlled by an individual LSTM. For the j-th agent, its policy takes the form of:

$$
\begin{aligned}
h_j^{t+1}, s_j^{t+1} &= LSTM(e(o_j^t), h_j^t, s_j^t) \\
a_j^t &= \pi(h_j^t),
\end{aligned}
$$

where $o_j^t$ is the observation of the j-th agent at time $t$, $e(\cdot)$ is an encoder function parameterized by a fully-connected neural network and $\pi$ is an agent's action policy. Also, $h_j^t$ and $s_j^t$ are the hidden and cell states of the LSTM. We use the same LSTM model for all agents, sharing their parameters. This way, the model is invariant to permutations of the agents.

IC3Net extends this independent controller model by allowing agents to communicate their internal state, gated by a discrete action. The policy of the j-th agent in a IC3Net is given by

$$
\begin{aligned}
g_j^{t+1} &= f^g(h_j^t) \\
h_j^{t+1}, s_j^{t+1} &= LSTM(e(o_j^t) + c_j^t, h_j^t, s_j^t) \\
c_j^{t+1} &= \frac{1}{J-1} C \sum_{j' \neq j} h_{j'}^{t+1} \odot g_{j'}^{t+1} \\
a_j^t &= \pi(h_j^t),
\end{aligned}
$$

where $c_j^t$ is the communication vector for the $j$-th agent, $C$ is a linear transformation matrix for transforming gated average hidden state to a communication tensor, $J$ is the number of alive agents currently present in the system and $f^g(.)$ is a simple network containing a soft-max layer for 2 actions (communicate or not) on top of a linear layer with non-linearity. The binary action $g_j^t$ specifies whether agent $j$ wants to communicate with others, and act as a gating function when calculating the communication vector. Note that the gating action for next time-step is calculated at current time-step. We train both the action policy $\pi$ and the gating function $f^g$ with REINFORCE (Williams, 1992).

In Sukhbaatar et al. (2016), individual networks controlling agents were interconnected, and they as a whole were considered as a single big neural network. This single big network controller approach required a definition of an unified loss function during training, thus making it impossible to train agents with different rewards.

In this work, however, we move away from the single big network controller approach. Instead, we consider multiple big networks with shared parameters each controlling a single agent separately. Each big network consists of multiple LSTM networks, each processing an observation of a single agent. However, only one of the LSTMs need to output an action because the big network is only controlling a single agent. Although this view has a little effect on the implementation (we can still use a single big network in practice), it allows us to train each agent to maximize its *individual* reward instead of a single global reward. This has two benefits: (i) it allows the model to be applied to both cooperative and competitive scenarios, (ii) it also helps resolve the credit assignment issue faced by many multi-agent (Sukhbaatar et al., 2016; Foerster et al., 2016a) algorithms while improving performance with scalability and is coherent with the findings in Chang et al. (2003).

## 4 EXPERIMENTS[1]

We study our network in multi-agent cooperative, mixed and competitive scenarios to understand its workings. We perform experiments to answer following questions:

1. Can our network learn the gating mechanism to communicate only when needed according to the given scenario? Essentially, is it possible to learn *when to communicate*?
2. Does our network using individual rewards scales better and faster than the baselines? This would clarify, whether or not, *individual rewards* perform better than *global rewards* in multi-agent communication based settings.

We first *analyze* gating action's ($g_t$) working. Later, we train our network in three chosen environments with variations in difficulty and coordination to ensure scalability and performance.

### 4.1 ENVIRONMENTS

We consider three environments for our analysis and experiments. (i) a predator-prey environment (**PP**) where predators with limited vision look for a prey on a square grid. (ii) a traffic junction environment (**TJ**) similar to Sukhbaatar et al. (2016) where agents with limited vision must learn to communicate in order to avoid collisions. (iii) StarCraft BroodWars[2] (**SC**) explore and combat

---

[1]The code is available at https://github.com/IC3Net/IC3Net.

[2]StarCraft is a trademark or registered trademark of Blizzard Entertainment, Inc., in the U.S. and/or other countries. Nothing in this paper should not be construed as approval, endorsement, or sponsorship by Blizzard Entertainment, Inc

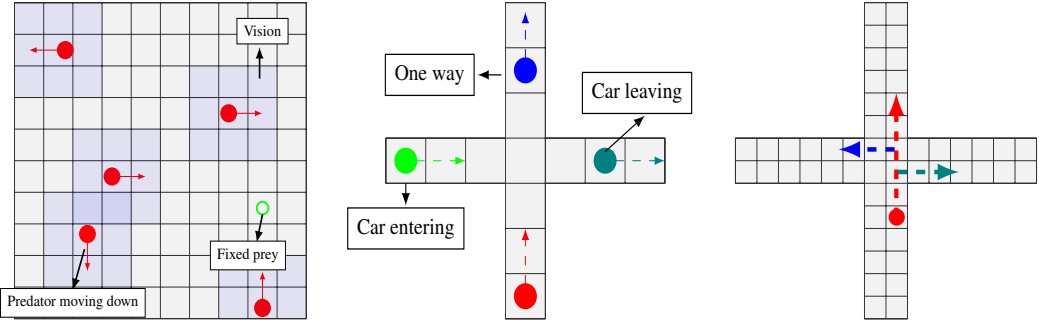

Figure 2: **Environments' Visualizations**. **(Left)** 10×10 version of predator-prey task where 5 predators(red circles) with limited vision of size 1 (blue region) try to catch a randomly initialized fixed prey (green circle). **(Center and Right)** Easy and medium versions of traffic junction task where cars have to cross the the whole path minimizing collisions using two actions, $gas$ and $brake$ respectively. Agents have zero vision and can only observe their own location. **(Right)** In medium version, chances of collision are increased due to more possible routes and increased number of cars.

tasks which test control on multiple agents in various scenarios where agent needs to understand and decouple observations for ***multiple opposing units***.

### 4.1.1 PREDATOR PREY

In this task, we have $n$ predators (agents) with limited vision trying to find a stationary prey. Once a predator reaches a prey, it stays there and always gets a positive reward, until end of episode (rest of the predators reach prey, or maximum number of steps). In case of zero vision, agents don't have a direct way of knowing prey's location unless they jump on it.

We design three cooperation settings (competitive, mixed and cooperative) for this task with different reward structures to test our network. See Appendix 6.3 for details on grid, reward structure, observation and action space. There is no loss or benefit from communicating in mixed scenario. In competitive setting, agents get lower rewards if other agents reach the prey and in cooperative setting, reward increases as more agents reach the prey. We compare with baselines using mixed settings in subsection 4.3.2 while explicitly learning and analyzing gating action's working in subsection 4.2.

We create three levels for this environment – as mentioned in Appendix 6.3 – to compare our network's performance with increasing number of agents and grid size. 10×10 grid version with 5 agents is shown in Figure 2 (left). All agents are randomly placed in the grid at start of an episode.

### 4.1.2 TRAFFIC JUNCTION

Following Sukhbaatar et al. (2016), we test our model on the traffic junction task as it is a good proxy for testing whether communication is working. This task also helps in supporting our claim that IC3Net provides good performance and faster convergence in fully-cooperative scenarios similar to mixed ones. In the traffic junction, cars enter a junction from all entry points with a probability $p_{arr}$. The maximum number of cars at any given time in the junction is limited. Cars can take two actions at each time-step, $gas$ and $brake$ respectively. The task has three difficulty levels (see Figure 2) which vary in the number of possible routes, entry points and junctions. We make this task harder by always setting vision to zero in all the three difficulty levels to ensure that task is not solvable without communication. See Appendix 6.4 for details on reward structure, observation and training.

### 4.1.3 STARCRAFT: BROODWARS

To fully understand the scalability of our architecture in more realistic and complex scenarios, we test it on StarCraft combat and exploration micro-management tasks in partially observable settings. StarCraft is a challenging environment for RL because it has a large observation-action space, many different unit types and stochasticity. We train our network on Combat and Explore task. The task's difficulty can be altered by changing the number of our units, enemy units and the map size.

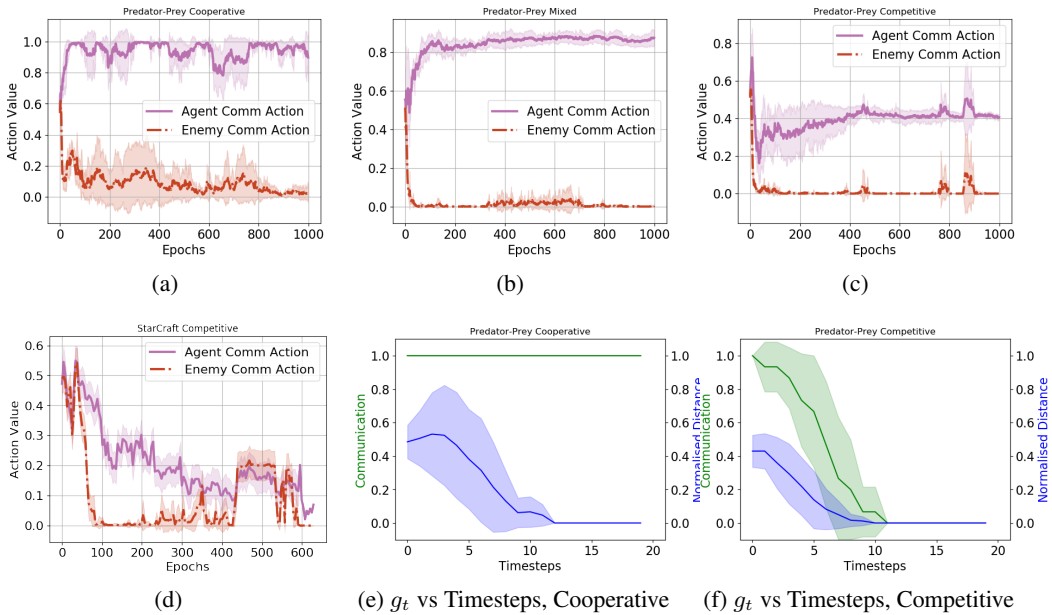

Figure 3: **Learning the Gating Action**: Plots show gating action $g_t$ for predators and prey averaged over each epoch in PP. In cooperative setting **(a, e)** agent almost always communicate to increase their own reward. In **(b)** mixed setting and **(c)** competitive setting, predators only communicate when necessary and profitable. As is evident from **(f)**, they stop communicating once they reach prey. In all cases, prey almost never communicates with predators as it is not profitable for it. Similarly, in competitive scenario **(d)** for SC, team agents learn to communicate only when necessary due to the division of reward when near enemy, while enemy agent learns not to communicate as in PP.

By default, the game has macro-actions which allow a player to directly target an enemy unit which makes player's unit find the best possible path using the game's in-built path-finding system, move towards the target and attack when it is in a range. However, we make the task harder by (i) removing macro-actions making exploration harder (ii) limiting vision making environment partially observable(iii) unlike previous works (Wender & Watson, 2012; Ontanón et al., 2013; Usunier et al., 2017; Peng et al., 2017), initializing enemy and our units at random locations in a fixed size square on the map, which makes it challenging to find enemy units. Refer to Appendix 6.5.1 for reward, action, observation and task details. We consider two types of tasks in StarCraft:

**Explore:** In this task, we have $n$ agents trying to explore the map and find an enemy unit. This is a direct scale-up of the PP but with more realistic and stochastic situations.

**Combat:** We test an agent's capability to execute a complex task of combat in StarCraft which require coordination between teammates, exploration of a terrain, understanding of enemy units and formalism of complex strategies. We specifically test a team of $n$ agents trying to find and kill a team of $m$ enemies in a partially observable environment similar to the explore task. The agents, with their limited vision, must find the enemy units and kill all of them to score a win. More information on reward structure, observation and setup can be found in Appendix 6.5.1 and 6.5.2.

### 4.2 ANALYSIS OF THE GATING MECHANISM

We analyze working of gating action ($g_t$) in IC3Net by using cooperative, competitive and mixed settings in Predator-Prey (4.1.1) and StarCraft explore tasks (4.1.3). However, this time the enemy unit (prey) shares parameters with the predators and is trained with them. All of the enemy unit's actions are *noop* which makes it stationary. The enemy unit gets a *positive reward* equivalent to $r_{time}$ = 0.05 per timestep until no predator/medic is captures it; after that it gets a reward of 0.

For 5×5 grid in PP task, Figure 3 shows gating action (averaged per epoch) in all scenarios for (i) communication between predator and (ii) communication between prey and predators. We also test

on 50×50 map size for competitive and cooperative StaraCraft explore task and found similar results (Fig. 3d). We can deduce following observations:

- As can be observed in Figure 3a, 3b, 3c and 3d, in all the four cases, the prey learns not to communicate. If the prey communicates, predators will reach it faster. Since it will get 0 reward when an agent comes near or on top of it, it doesn't communicate to achieve higher rewards.

- In cooperative setting (Figure 3a, 3e), the predators are openly communicating with $g$ close to 1. Even though the prey communicates with the predators at the start, it eventually learns not to communicate; so as not to share its location. As all agents are communicating in this setting, it takes more training time to adjust prey's weights towards silence. Our preliminary tests suggest that in cooperative settings, it is beneficial to fix the gating action to 1.0 as communication is almost always needed and it helps in faster training by skipping the need to train the gating action.

- In the mixed setting (Figure 3b), agents don't always communicate which corresponds to the fact that there is no benefit or loss by communicating in mixed scenario. The prey is easily able to learn not to communicate as the weights for predators are also adjusted towards non-cooperation from the start itself.

- As expected due to competition, predators rarely communicate in competitive setting (Figure 3c, 3d). Note that, this setting is not fully-adversarial as predators can initially explore faster if they communicate which can eventually lead to overall higher rewards. This can be observed as the agents only communicate while it's profitable for them, i.e. before reaching the prey (Figure 3f) as communicating afterwards can impact their future rewards.

Experiments in this section, empirically suggest that agents can **"learn to communicate when it is profitable"**; thus allowing same network to be used in all settings.

### 4.3 Scalability and Generalization Experiments

In this section, we look at bigger versions of our environments to understand scalability and generalization aspects of IC3Net.

#### 4.3.1 Baselines

For **training details**, refer to Appendix 6.1. We compare IC3Net with **baselines** specified below in all scenarios.

**Individual Reward Independent Controller (IRIC):** In this controller, model is applied individually to all of the agents' observations to produce the action to be taken. Essentially, this can be seen as IC3Net without any communication between agents; but with individualized reward for each agent. Note that no communication makes gating action ($g_t$) *ineffective*.

**Independent Controller (IC - IC3Net w/o Comm and IR):** Like IRIC except the agents are trained with a global average reward instead of individual rewards. This will help us understand the credit assignment issue prevalent in CommNet.

**CommNet:** Introduced in Sukhbaatar et al. (2016), CommNet allows communication between agents over a channel where an agent is provided with the average of hidden state representations of other agents as a communication signal. Like IC3Net, CommNet also uses continuous signals to communicate between the agents. Thus, CommNet can be considered as IC3Net without both the gating action ($g_t$) and individualized rewards.

#### 4.3.2 Results

We discuss major results for our experiments in this section and analyze particular behaviors/patterns of agents in Appendix 6.2.

**Predator Prey:** Table 1 (left) shows average steps taken by the models to complete an episode i.e. find the prey in mixed setting (we found similar results for cooperative setting shown in appendix). IC3Net reaches prey faster than the baselines as we increase the number of agents as well as the size of the maze. In 20×20 version, the gap in average steps is almost 24 steps, which is a substantial improvement over baselines. Figure 4 (right) shows the scalability graph for IC3Net and CommNet which supports the claim that with the increasing number of agents, IC3Net converges faster at a

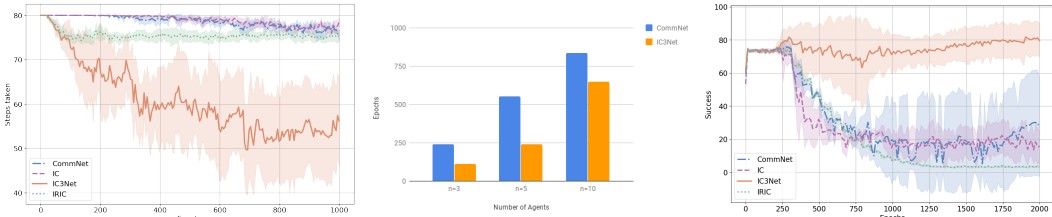

Figure 4: **Result Plots for PP and TJ task. (Left)** Average steps taken to complete an episode in 20×20 grid. **(Center)** IC3Net converges faster than CommNet as the number of predators (agents) increase in Predator-Prey environment. **(Right)** Success % in medium TJ task trained with curriculum. Performance and convergence of IC3Net is superior than baselines.

| | Predatory-Prey Mixed (Avg. Steps) | | | | Traffic Junction (Success %) | | |
|---|---|---|---|---|---|---|---|
| **Model** | **5x5, n=3** | **10x10, n=5** | **20x20, n=10** | **Model** | **Easy** | **Medium** | **Hard** |
| IRIC | 16.5± 0.1 | 28.1± 0.2 | 75.0± 1.4 | IRIC | 29.8± 0.7 | 3.4± 0.5 | 35.0± 0.6 |
| IC | 16.4± 0.49 | 28.0± 0.74 | 77.4± 0.8 | IC | 30.2± 0.4 | 3.4± 0.5 | 47.0± 2.9 |
| CommNet | 9.1± 0.1 | 13.1± 0.01 | 76.5± 1.3 | CommNet | 93.0± 4.2 | 54.3± 14.2 | 50.2± 3.5 |
| IC3Net | **8.9± 0.02** | 13.0± 0.02 | **52.4± 3.4** | IC3Net | 93.0± 3.7 | **89.3± 2.5** | **72.4± 9.6** |

Table 1: **Predator Prey** (Left): Avg. number of steps taken to complete the episode in three different environment sizes in mixed settings. IC3Net completes the episode faster than the baselines by finding the prey. **Traffic Junction** (Right): Success rate on various difficulty levels with zero vision for all. IC3Net provides better performance than baselines consistently especially as the scale increases.

better optimum than CommNet. Through these results on the PP task, we can see that compared to IC3Net, CommNet doesn't work well in mixed scenarios. Finally, Figure 4 (left) shows the training plot of 20×20 grid with 10 agents trying to find a prey. The plot clearly shows the faster performance improvement of IC3Net in contrast to CommNet which takes long time to achieve a minor jump. We also find same pattern of the gating action values as in 4.2.

**Traffic Junction:** Table 1 (right) shows the success ratio for traffic junction. We fixed the gating action to 1 for TJ as discussed in 4.2. With zero vision, it is not possible to perform well without communication as evident by the results of IRIC and IC. Interestingly, IC performs better than IRIC in the hard case, as we believe without communication, the global reward in TJ acts as a better indicator of the overall performance. On the other hand, with communication and better knowledge of others, the global reward training face a credit assignment issue which is alleviated by IC3Net as evident by its superior performance compared to CommNet. In Sukhbaatar et al. (2016), well-performing agents in the medium and hard versions had vision > 0. With zero vision, IC3Net is to CommNet and IRIC with a performance gap greater than 30%. This verifies that individualized rewards in IC3Net help achieve a better or similar performance than CommNet in fully-cooperative tasks with communication due to a better credit assignment.

**StarCraft:** Table 2 displays win % and the average number of steps taken to complete an episode in StarCraft explore and combat tasks. We specifically test on (i) Explore task: 10 medics finding 1 enemy medic on 50×50 cell grid (ii) On 75×75 cell grid (iii) Combat task: 10 Marines vs 3 Zealots on 50 x 50 cell grid. Maximum steps in an episode are set to 60. The results on the explore task are similar to Predator-Prey as IC3Net outperforms the baselines. Moving to a bigger map size, we still see the performance gap even though performance drops for all the models.

On the combat task, IC3Net performs comparably well to CommNet. A detailed analysis on IC3Net's performance in StarCraft tasks is provided in Appendix 6.2.1. To confirm that 10 marines vs 3 zealots is hard to win, we run an experi-

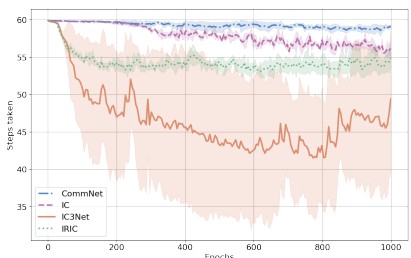

Figure 5: **Average steps taken** to complete an episode of StarCraft Explore-10 Medic 50×50 task.

| StarCraft task | IRIC | | IC | | CommNet | | IC3Net | |
|---|---|---|---|---|---|---|---|---|
| | Win % | Steps | Win % | Steps | Win % | Steps | Win % | Steps |
| Exp-10M $50 \times 50$ | 35.4± 1.7 | 52.7± 0.6 | 18.0± 1.63 | 55.5± 0.4 | 9.2± 3.12 | 58.4± 0.5 | **64.2**± **17.7** | **41.2**± **8.4** |
| Exp-10M $75 \times 75$ | 9.0± 4.2 | 58.6± 0.9 | 1.0± 0.2 | 59.1± 0.4 | 2.1± 0.3 | 59.9± 0.1 | **17.0**± **15.2** | **57.2**± **3.5** |
| Cbt-10Mv3Ze | 74.6± 5.1 | 35.0± 0.8 | 51.8± 3.0 | 48.6± 0.4 | 88.0± 7.2 | 33.3± 1.2 | 87.4± 1.0 | 33.6± 0.2 |

Table 2: **StarCraft Results**: Win Ratio and average number of steps taken to complete episodes for explore (Exp) tasks with 10 Marines (M) on different grid sizes ($50 \times 50$ and $75 \times 75$) and a combat (Cbt) task with 10 Marines (M) vs 3 Zealots (Ze) on a grid of size $50 \times 50$. IC3Net beats the baselines with huge margin in case of exploration tasks, while it is as good as CommNet in case of 10 Marines vs 3 Zealots combat task.

ment on reverse scenario where our agents control 3 Zealots initialized separately and enemies are 10 marines initialized together. We find that both IRIC and IC3Net reach a success percentage of 100% easily. We find that even in this case, IC3Net converges faster than IRIC.

## 5 CONCLUSIONS AND FUTURE WORK

In this work, we introduced IC3Net which aims to solve multi-agent tasks in various cooperation settings by learning when to communicate. Its continuous communication enables efficient training by backpropagation, while the discrete gating trained by reinforcement learning along with individual rewards allows it to be used in all scenarios and on larger scale.

Through our experiments, we show that IC3Net performs well in cooperative, mixed or competitive settings and learns to communicate only when necessary. Further, we show that agents learn to stop communication in competitive cases. We show scalability of our network by further experiments. In future, we would like to explore possibility of having multi-channel communication where agents can decide on which channel they want to put their information similar to communication groups but dynamic. It would be interesting to provide agents a choice of whether to listen to communication from a channel or not.

**Acknowledgements** Authors would like to thank Zeming Lin for his consistent support and suggestions around StarCraft and TorchCraft.

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

# 6 APPENDIX

## 6.1 TRAINING DETAILS

We set the hidden layer size to 128 units and we use LSTM (Hochreiter & Schmidhuber, 1997) with recurrence for all of the baselines and IC3Net. We use RMSProp (Tieleman & Hinton, 2012) with initial learning rate as a tuned hyper-parameter. All of the models use skip-connections (He et al., 2016). The training is distributed over 16 cores and each core runs a mini-batch till total episodes steps are 500 or more. We do 10 weight updates per epoch. We run predator-prey, StarCraft experiments for 1000 epochs, traffic junction experiment for 2000 epochs and report the final results. In mixed case, we report the mean score of all agents, while in cooperative case we report any agent's score as they are same. We implement our model using PyTorch and environments using Gym (Brockman et al., 2016).We use REINFORCE (Williams, 1992) to train our setup. We conduct 5 runs on each of the tasks to compile our results. The training time for different tasks varies; StarCraft tasks usually takes more than a day (depends on number of agents and enemies), while predator-prey and traffic junction tasks complete under 12 hours.

## 6.2 RESULTS ANALYSIS

In this section, we analyze and discuss behaviors/patterns in the results on our experiments.

### 6.2.1 IC3NET IN STARCRAFT-COMBAT TASK

As observed in Table 2, IC3Net performs better than CommNet in explore task but doesn't outperform it on Combat task. Our experiments and visualizations of actual strategy suggested that compared to exploration, combat can be solved far easily if the units learn to stay together. Focused firepower with more attack quantity in general results in quite good results on combat. We verify this hypothesis by running a heuristics baseline "attack closest" in which agents have full vision on map and have macro actions available[3]. By attacking the closest available enemy together the agents are able to kill zealots with success ratio of $76.6_{\pm 8}$ calculated over 5 runs, even though initialized separately. Also, as described in Appendix 6.5.2, the global reward in case of win in Combat task is relatively huge compared to the individual rewards for killing other units. We believe that with coordination to stay together, huge global rewards and focus fire–which is achievable through simple cooperation–add up to CommNet's performance in this task.

Further, in exploration we have seen that agents go in separate direction and have individual rewards/sense of exploration which usually leads to faster exploration of an unexplored area. Thinking in simple terms, exploration of an house would be faster if different people handle different rooms. Achieving this is hard in CommNet because global rewards don't exactly tell your individual contributions if you had explored separately. Also in CommNet, we have observed that agents follow a pattern where they get together at a point and explore together from that point which further signals that using CommNet, it is easy to get together for agents[4].

### 6.2.2 VARIANCE IN IC3NET

In Figure 5, we have observed significant variance in IC3Net results for StarCraft. We performed a lot of experiments on StarCraft and can attribute the significant variance to stochasticity in the environment. There are a huge number of possible states in which agents can end up due to millions of possible interactions and their results in StarCraft. We believe it is hard to learn each one of them. This stochasticity variance can even be seen in simple heuristics baselines like "attack closest" (6.2.1) and is in-fact an indicator of how difficult is it to learn real-world scenarios which also have

---

[3]Macro-actions corresponds to "right click" feature in StarCraft and Dota in which a unit can be called to attack on other unit where units follows the shortest path on map towards the unit to be attacked and once reached starts attacking automatically, this essentially overpowers "attack closest" baseline to easily attack anyone under full-vision without any exploration.

[4]You can observe the above stated pattern for CommNet in PP in this video: https://gfycat.com/IllustriousMarvelousKagu. This video has been generated using trained CommNet model on PP-Hard. Here Red 'X' are predators and 'P' is the prey to be found. We can observe the pattern where the agents get together to find the prey leading to slack eventually

same amount of stochasticity. We believe that we don't see similar variance in CommNet and other baselines because adding gating action increases the action-state-space combinations which yields better results while being difficult to learn sometimes. Further, this variance is only observed in higher Win % models which requires to learn more state spaces.

### 6.2.3 COMMNET IN STARCRAFT-EXPLORE TASKS

In Table 2, we can observe that CommNet performs worse than IRIC and IC in case of StarCraft-Explore task. In this section, we provide a hypothesis for this result. First, we need to notice is that IRIC is better than IC also overall, which points to the fact that individualized reward are better than global rewards in case of exploration. This makes sense because if agents cover more area and know how much they covered through their own contribution (individual reward), it should lead to overall more coverage, compared to global rewards where agents can't figure out their own coverage but instead overall one. Second, in case of CommNet, it is easy to communicate and get together. We observe this pattern in CommNet[4] where agents first get together at a point and then start exploring from there which leads to slow exploration, but IC is better in this respect because it is hard to gather at single point which inherently leads to faster exploration than CommNet. Third, the reward structure in the case of mixed scenario doesn't appreciate searching together which is not directly visible to CommNet and IC due to global rewards.

## 6.3 DETAILS OF PREDATOR PREY

In all the three settings, cooperative, competitive and mixed, a predator agent gets a constant time-step penalty $r_{explore} = -0.05$, until it reaches the prey. This makes sure that agent doesn't slack in finding the prey. In the mixed setting, once an agent reaches the prey, the agent always gets a positive reward $r_{prey} = 0.05$ which doesn't depend on the number of agents on prey. . Similarly, in the cooperative setting, an agent gets a positive reward of $r_{coop} = r_{prey} * n$, and in the competitive setting, an agent gets a positive reward of $r_{comp} = r_{prey} / n$ after it reaches the prey, where $n$ is the number of agents on the prey. The total reward at time $t$ for an agent $i$ can be written as:

$$r_i^{pp}(t) = \delta_i * r_{explore} + (1 - \delta_i) * n_t^\lambda * r_{prey} * |\lambda|$$

where $\delta_i$ denotes whether agent $i$ has found the prey or not, $n_t$ is number of agents on prey at time-step $t$ and $\lambda$ is -1, 0 and 1 in the competitive, mixed and cooperative scenarios respectively. Maximum episode steps are set to 20, 40 and 80 for 5×5, 10×10 and 20×20 grids respectively. The number of predators are 5, 10 and 20 in 5×5, 10×10 and 20×20 grids respectively. Each predator can take one of the five basic movement actions i.e. $up, down, left, right$ or $stay$. Predator, prey and all locations on grid are considered unique classes in vocabulary and are represented as one-hot binary vectors. Observation $obs$, at each point will be the sum of all one-hot binary vectors of location, predators and prey present at that point. With vision of 1, observation of each agent have dimension $3^2 \times |obs|$.

### 6.3.1 EXTRA EXPERIMENTS

| | Predator Prey Cooperative (Avg. Rewards) | | |
|---|---|---|---|
| Model | 5x5, n=3 | 10x10, n=5 | 20x20, n=10 |
| IRIC | 0.48± 0.001 | 4.08± 0.049 | 5.77± 0.130 |
| IC | 0.47± 0.009 | 3.78± 0.082 | 4.99± 0.529 |
| CommNet | 1.56± 0.010 | 6.94± 0.020 | 19.99± 0.62 |
| IC3Net | 1.57± 0.008 | 6.85± 0.144 | **21.09± 0.579** |

Table 3: **Predator-Prey Cooperative**: Avg. rewards in three difficulty levels of predator-prey environment in cooperative setting. IC3Net performs equivalently or better than baselines consistently.

Table 3 shows the results for IC3Net and baselines in the cooperative scenario for the predator-prey environment. As the cooperative reward function provides more reward after a predator reaches the prey, the comparison is provided for rewards instead of average number of steps. IC3Net performs better or equal to CommNet and other baselines in all three difficulty levels. The performance gap closes in and increases as we move towards bigger grids which shows that IC3Net is more scalable

due to individualized rewards. More importantly, even with the extra gating action training, IC3Net can perform comparably to CommNet which is designed for cooperative scenarios which suggests that IC3Net is a suitable choice for all cooperation settings.

To analyze the effect of gating action on rewards in case of mixed scenario where individualized rewards alone can help a lot, we test Predator Prey mixed cooperation setting on 20x20 grid on a baseline in which we set gating action to 1 (global communication) and uses individual rewards (IC2Net/CommNet + IR). We find average max steps to be $50.24_{\pm 3.4}$ which is lower than IC3Net. This means that (i) individualized rewards help a lot in mixed scenarios by allowing agents to understand there contributions (ii) adding the gating action in this case has an overhead but allows the same model to work in all settings (even competitive) by *"learning to communicate"* which is more close to real-world humans with a negligible hit on the performance.

## 6.4 DETAILS OF TRAFFIC JUNCTION

Traffic junction's observation vocabulary has one-hot vectors for all locations in the grid and car class. Each agent observes its previous action, route identifier and a vector specifying sum of one-hot vectors for all classes present at that agent's location. Collision occurs when two cars are on same location. We set maximum number of steps to 20, 40 and 60 in easy, medium and hard difficulty respectively. Similar to Sukhbaatar et al. (2016), we provide a negative reward $r_{coll}$ = -10 on collision. To cut off traffic jams, we provide a negative reward $\tau_i r_{time}$ = -0.01 $\tau_i$ where $\tau_i$ is time spent by the agent in the junction at time-step $t$. Reward for $i$th agent which is having $C_i^t$ collisions at time-step $t$ can be written as:

$$r_i^{tj}(t) = r_{coll}C_i^t + r_{time}\tau_i$$

We utilized curriculum learning Bengio et al. (2009) to make the training process easier. The $p_{arrive}$ is kept at the $start$ value till the first 250 epochs and is then linearly increased till the $end$ value during the course from 250th to 1250th epoch. The $start$ and $end$ values of $p_{arrive}$ for different difficulty levels are indicated in Table 4. Finally, training continues for another 750 epochs. The learning rate is fixed at 0.003 throughout. We also implemented three difficulty variations of the game explained as follows.

| Difficulty | P-arrive Start | End | N-total | Arrival Points | Routes per Entry Point | Two-Way | Junctions | Dimension |
|---|---|---|---|---|---|---|---|---|
| Easy | 0.1 | 0.3 | 5 | 2 | 1 | F | 1 | 7×7 |
| Medium | 0.05 | 0.2 | 10 | 4 | 3 | T | 1 | 14×14 |
| Hard | 0.02 | 0.05 | 20 | 8 | 7 | T | 4 | 18×18 |

Table 4: **Traffic Junction**: Variations in different traffic junction difficulty levels. T refers to True as in the difficulty level has 2-way roads and F refers to False as in the difficulty level has 1-way roads.

The easy version is a junction of two one-way roads on a $7 \times 7$ grid. There are two arrival points, each with two possible routes and with a $N_{total}$ value of 5.

The medium version consists of two connected junctions of two-way roads in $14 \times 14$ as shown in Figure 2 (right). There are 4 arrival points and 3 different routes for each arrival point and have $N_{total} = 20$.

The harder version consists of four connected junctions of two-way roads in $18 \times 18$ as shown in Figure 6. There are 8 arrival points and 7 different routes for each arrival point and have $N_{total} = 20$.

### 6.4.1 IRIC AND IC PERFORMANCE

In Table 1, we notice that IRIC and IC perform worst in medium level compared to the hard level. Our visualizations suggest that this is due to high final add-rate in case of medium version compared to hard version. Collisions happen much more often in medium version leading to less success rate (an episode is considered failure if a collision happens) compared to hard where initial add-rate is low to accommodate curriculum learning for hard version's big grid size. The final add-rate in case of hard level is comparatively low to make sure that it is possible to pass a junction without a collision as with more entry points it is easy to collide even with a small add-rate.



Figure 6: **Hard difficulty level of traffic junction task**. Level has four connected junctions, eight entry points and at each entry point there are 7 possible routes increasing chances of a collision. We use curriculum learning to successfully train our models on hard level.

## 6.5 STARCRAFT DETAILS

### 6.5.1 OBSERVATION AND ACTIONS

**Explore:** To complete the explore task, agents must be within a particular range of enemy unit called *explore vision*. Once an agent is within *explore vision* of enemy unit, we *noop* further actions. The reward structure is same as the PP task with only difference being that an agent needs to be within the *explore vision* range of the enemy unit instead of being on same location to get a non-negative reward. We use *medic* units which don't attack enemy units. This ensures that we can simulate our explore task without any of kind of combat happening and interfering with the goal of the task. Observation for each agent is its *(absolute x, absolute y)* and enemy's *(relative x, relative y, visible)* where *visible*, *relative x* and *relative y* are 0 when enemy is not in *explore vision* range. Agents have 9 actions to choose from which includes 8 basic directions and one stay action.

**Combat:** Agent observes its own *(absolute x, absolute y, healthpoints + shield, weapon cooldown, previous action)* and *(relative x, relative y, visible, healthpoints + shield, weapon cooldown)* for each of the enemies. *relative x* and *relative y* are only observed when enemy is visible which is corresponded by *visible* flag. All of the observations are normalized to be in between (0, 1). Agent has to choose from $9 + m$ actions which include 9 basic actions and 1 action for attacking each of the $m$ agents. Attack actions only work when the enemy is within the *sight* range of the agent, otherwise it is a *noop*. In combat, we don't compare with prior work on StarCraft because our environment setting is much harder, restrictive, new and different, thus, not directly comparable.

### 6.5.2 COMBAT REWARD

To avoid slack in finding the enemy team, we provide a negative reward $r_{time}$ = -0.01 at each timestep when the agent is not involved in a combat. At each timestep, an agent gets as reward the difference between (i) its normalized health in current and previous timestep (ii) normalized health at previous timestep and current timestep for each of the enemies it has attacked till now. At the end of the episode, terminal reward for each agents consists of (i) all its remaining health * 3 as negative reward (ii) 5 * $m$ + all its remaining health * 3 as positive reward if agents win (iii) normalized remaining health * 3 for all of the alive enemies as negative reward on lose. In this task, the group of enemies is initialized together randomly in one half of the map and our agents are initialized separately in other half which makes task even harder, thus requiring communication. For an automatic way of individualizing rewards, please refer to Foerster et al. (2018).

### 6.5.3 EXAMPLE SEQUENCE OF STATES IN COOPERATIVE EXPLORE MODE

We provide an example sequence of states in StarCraft cooperative explore mode in Figure 7. As soon as one of the agents finds the enemy unit, the other agents get the information about enemy's location through communication and are able to reach it faster.

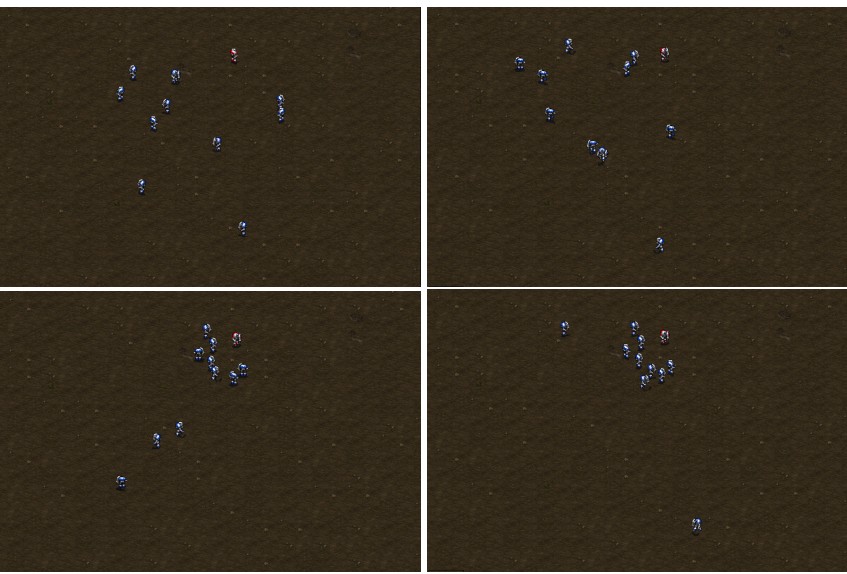

Figure 7: State sequence in SC Cooperative Explore. We can see how agents which are randomly initialized communicate to reach the enemy faster. Once, some agents are near the prey and other reach very fast due to communication

