# OpenReview forum: "Learning when to Communicate at Scale in Multiagent Cooperative and Competitive Tasks"
_ICLR.cc/2019/Conference_

### Official Review · AnonReviewer2 · 2018-11-03
**very well written paper, good experiment section, method of communication can be more motivated**

**Rating:** 6
**Confidence:** 3

**Review:**

This work is an extension to the work of Sukbaatar et al. (2016) with two main differences:
1) Selective communication: agents are able to decide whether they want to communicate.
2) Individualized reward: Agents receive individual rewards; therefore, agents are aware of their contribution towards the goal.
These two new extensions enable their model to work in either cooperative or a mix of competitive and competitive/collaborative settings. The authors also claim these two extensions enable their model to converge faster and better.
The paper is well written, easy to follow, and everything has been explained quite well. The experiments are competent in the sense that the authors ran their model in four different environments (predator and prey, traffic junction, StarCraft explore, and StarCraft combat). The comparison between their model with three baselines was extensive; they reported the mean and variance over different runs. I have some concerns regarding their method and the experiments which are brought up in the following:

Method:

In a non-fully-cooperative environment, sharing hidden state entirely as the only option for communicate is not very reasonable; I think something like sending a message is a better option and  more realistic (e.g., something like the work of Mordatch & Abbeel, 2017)

Experiment:

The experiment "StarCraft explore" is similar to predator-prey; therefore, instead of explaining StarCraft explore, I would like to see how the model works in StarCraft combat. Right now, the authors explain a bit about the model performance in Starcraft combat, but I found the explanation confusing.

Authors provide 3 baselines:
1) no communication, but IR
2) no communication, no IR
3) global communication, no IR (commNet)

I think having a baseline that has global communication with IR can show the effect of selective communication better.

There are some questions in the experiment section that have not been addressed very well. For example:
 Is there any difference between the results of table 1, if we look at the cooperative setup?
Does their model outperform a model which has global communication with IR?
Why do IRIC and IC work worst in the medium in comparison to hard in TJ in table1?
Why is CommNet work worse than IRIC and IC in table 2?

---

> ### Author Response · Authors · 2018-11-26
> **Thank you for your detailed comments, suggestions and feedback (2/2)**
>
> 5. Why do IRIC and IC work worst in the medium in comparison to hard in TJ in table1?
> Our visualizations suggest that this is due to high final add-rate in case of medium version compared to hard version. Collisions happen much more often in medium version leading to less success rate (an episode is considered failure if a collision happens) compared to hard where initial add-rate is low to accommodate curriculum learning for hard version’s big grid size. The final add-rate in case of hard level is comparatively low to make sure that it is possible to pass a junction without a collision as with more entry points it is easy to collide even with a small add-rate.
>
> 6. Why is CommNet work worse than IRIC and IC in table 2?
> First, we need to notice is that IRIC is better than IC also overall, which points to the fact that individualized reward are better than global rewards in case of exploration. This makes sense because if agents cover more area and know how much they covered through their own contribution (individual reward), it should lead to overall more coverage, compared to global rewards where agents can’t figure out their own coverage but
> instead overall one. Second, in case of CommNet, it is easy to communicate and get together. We observe this pattern in CommNet where agents first get together at a point and then start exploring from there which leads to slow exploration, but IC is better in this respect because it is hard to gather at single point which inherently leads to faster exploration than CommNet..  Third, the reward structure in mixed scenario doesn’t appreciate searching together which is not directly visible to CommNet and IC due to global rewards.
>
> Note: You can observe the pattern for CommNet we just talked about at https://gfycat.com/IllustriousMarvelousKagu . This video has been generated using trained CommNet model on PP-Hard. Red ‘X’ are predators and ‘P’ is the prey to be found. We can observe the pattern where the agents get together to find the prey leading to slack eventually.
>
> We have updated our paper to reflect the answers in Appendix. Thanks for providing us a detailed review and insightful questions.

---

> ### Author Response · Authors · 2018-11-26
> **Thank you for your detailed comments, suggestions and feedback (1/2)**
>
> Dear Reviewer,
>
> We thank you for your insightful and helpful comments. We have responded to your comments, suggestions and concerns inline below. We hope our answers clarify your concerns.
>
> 1.In a non-fully-cooperative environment, sharing hidden state entirely as the only option for communicate is not very reasonable; I think something like sending a message is a better option and more realistic (e.g., something like the work of Mordatch & Abbeel, 2017)
>
> We agree with the reviewer that sharing hidden state entirely is not the only option for communication but we will like to point out that it is a very reasonable one. We build this claim on the fact that using fixed vocabularies is not scalable to diverse scenarios such as StarCraft with a lot of actions. We believe that vocabulary of agents should be as vast as humans as should be able to capture a lot of state-space scenarios. Even though discrete messages are more interpretable, as we increase the vocabulary size to make it more vast, it gets harder to train agent to learn all the symbols using RL which in-effect limits the scalability of the approach. Since, hidden state is direct encoding of observations in latent space, it effectively capture most of the information necessary in encoded form and thus, create a version of vast vocabulary which is easily trainable.
>
> 2. The experiment "StarCraft explore" is similar to predator-prey; therefore, instead of explaining StarCraft explore, I would like to see how the model works in StarCraft combat. Right now, the authors explain a bit about the model performance in Starcraft combat, but I found the explanation confusing.
>
> Due to space constraints, we pushed out intrinsic and trivial details of StarCraft Combat task into Section 6.4 (Appendix). We have revised our text to reduce the details on StarCraft Explore and added reference to proper appendices for Combat task. We updated the paper to provide detailed  explanation about the performance of the model in Section 6.2.1.
>
> 3. Is there any difference between the results of table 1, if we look at the cooperative setup?
>
> We have added Table 3 in Section 6.2.1 which shows results of Table 1 in cooperative setup. IC3Net performs equivalently or better than the baselines which shows that IC3Net is a suitable choice for any cooperation setting. As the difficulty level increases the performance gap between IC3Net and baselines increases. IC3Net is able to perform comparable and better than CommNet which is designed for cooperative scenarios and works well in them which suggests that agents can learn to communicate in cooperative scenarios.
>
> 4. I think having a baseline that has global communication with IR can show the effect of selective communication better. Does their model outperform a model which has global communication with IR?
>
> We would like to note that a baseline that has global communication with IR will essentially point to IC3Net with gating action always set to 1.  As our experiments had suggested and we had observed, communication action is learnable but adds extra layer of unnecessary dependency in case of cooperative scenarios. That’s why we suggest users to use communication action to 1 and increase training speed in case of cooperative scenarios.
>
> This might even work in case of  mixed scenarios if the competitive component is not as strong as cooperative component. In competitive scenarios, such a baseline doesn’t make sense and doesn’t fit in the structure of the task as it is useless to share everything with the opponent. So, we add results on PP-mixed hard version to Appendix 6.3.1 when communication action is set to 1. We would also like to reiterate that these results can be better or equivalent to IC3Net as learning the gating action (even though learnable) can add extra complexity overhead (as observed in the result). At the same time, gating action allows the same model to work in all scenarios regardless of the cooperation setting which is not possible with global communication.

---

> ### Author Response · Authors · 2018-11-30
> **Request for feedback**
>
> Hi Reviewer2,
> Thanks once again for the helpful feedback on our work. Following our response and updated paper, if you have any updated questions/feedback/thoughts, we would be very happy to address them. We have open sourced our code at https://github.com/IC3Net/IC3Net . Thanks :) .

---

### Official Review · AnonReviewer1 · 2018-11-04
**In this work, the authors propose an interesting gating scheme allowing agents to communicate in an multi-agent RL setting.**

**Rating:** 6
**Confidence:** 3

**Review:**

From a methodological perspective, this paper describes a simple bu clever learning architecture with individual agents able to decide when to communicate through a learned gating mechanism. Each agent is an LSTM able to decide at each time point which aspects of its internal state should be exposed to other agents through this gating mechanism. The presentation of this method is clear to a level that should allows the reader to implement this him/herself. It would be great if the code associated to this could be released but the presentation allows for reproducibility.

The experiments are interesting as well. Experimental results are presented on 3 problems and compared with known baselines from the academic community. The obtained results do show the merit of the approach. That being said, while the experimental results are extensive, there are places that could benefit from more clarity. For instance, I have found section 4.2 a bit dry. For instance, I had to read the plots caption and the text several times to map get at the deductions made in 4.2. Given the importance of gating in this work, I recommend expanding on this a bit (if space allows it). Small note: in the caption for Figure 3, on the fourth line, did you mean (f) instead of (d) when arguing that agents stop communicating once they reach the prey ( or am I missing something here)? Also, would it be possible to provide more insights on why IC3Net is doing better than CommNet except for the Combat-10Mv3Ze task (last table before the conclusion, what makes this task harder for IC3Net)? Another observation is on the variance terms that are reported for IC3Net. They are often (not always but definitely in the last table before the conclusion) quite higher when compared to the values associated with the baselines. Can this be explained? Another small thing: please add captions to your tables (at least a table number; I think that Table 2 does not have a caption).


Overall, the paper is well written, interesting. Addressing the questions raised above would definitely help me and probably the eventual readers better appreciate its quality.

---

> ### Author Response · Authors · 2018-11-26
> **Thank you for your detailed comments and feedback (2/2)**
>
> 6. Another observation is on the variance terms that are reported for IC3Net. They are often (not always but definitely in the last table before the conclusion) quite higher when compared to the values associated with the baselines. Can this be explained?
>
> We agree with the  reviewer about the significant variance in case of IC3Net for StarCraft. We have performed a lot of experiments on StarCraft and can attribute the significant variance to stochasticity in the environment. There are a huge number of possible states in which agents can end up due to millions of possible interactions and their results in StarCraft and we believe it is hard to learn each one of them. This stochasticity variance can even be seen in simple heuristics baselines like “attack closest”  and is in-fact an indicator of how difficult is it to learn real-world scenarios which also have same amount of stochasticity.
>
> Albeit, one can ask why don’t we see similar variance in CommNet and others. We believe that this might be due to the fact that adding gating action increases the action-state-space combinations which yields better results while being difficult to learn sometimes. Other point to be noted is that this variance is generally seen when the Win % (requires learning more states) is above some particular threshold which is close to  nothing in baselines. We have added a section 6.2.2 on variance to the paper.
>
> 7. Another small thing: please add captions to your tables (at least a table number; I think that Table 2 does not have a caption).
>
> We have updated the paper to better reflect captions for the tables. Thanks for pointing this out.

---

> ### Author Response · Authors · 2018-11-26
> **Thank you for your detailed comments and feedback (1/2)**
>
> Dear Reviewer,
>
> We thank you for your insightful and helpful comments. We have responded to your comments, suggestions and concerns inline below. We hope our answers clarify your concerns.
>
> 1.  It would be great if the code associated to this could be released but the presentation allows for reproducibility.
>
> As requested and promised, code will be available with a simple starter README at the link mentioned in the paper (https://github.com/IC3Net/IC3Net ) soon. We will update the thread when we do so.
>
> 2. I have found section 4.2 a bit dry. For instance, I had to read the plots caption and the text several times to map get at the deductions made in 4.2.
>
> We have modified and made Section 4.2 more coherent so that it can be followed easily.
>
> 3. Given the importance of gating in this work, I recommend expanding on this a bit (if space allows it).
>
> We have decided to expand on gating in the camera-ready version as it will have more space. Further, we would like to take this moment to also mention that while gating is important, our work’s other major contribution is showing how well individualized rewards can work.
>
> 4. Small note: in the caption for Figure 3, on the fourth line, did you mean (f) instead of (d) when arguing that agents stop communicating once they reach the prey ( or am I missing something here)?
>
> Thanks for notifying about this. You are correct; we did mean (f) there. We have fixed this in the updated version of the paper.
>
> 5. Also, would it be possible to provide more insights on why IC3Net is doing better than CommNet except for the Combat-10Mv3Ze task (last table before the conclusion, what makes this task harder for IC3Net)?
>
> Our experiments and visualizations of actual strategy suggested that compared to exploration, combat can be solved far easily if the units learn to stay together. Focused firepower with more quantity in general results in quite good results on combat.  We verify this hypothesis by running a heuristics baseline “attack closest” in which agents have full vision on map and have macro actions available. By attacking the closest available enemy together the agents are able to kill zealots with success ratio of 76.6 +- 8 calculated over 5 runs  even though initialized separately. Also, as described in Section 6.5.2, the global reward in case of win in Combat task is relatively huge compared to the individual rewards for killing other units. We believe that with coordination to stay together, huge global rewards and focus fire--which is achievable through simple cooperation--add up to CommNet's performance in this task.
>
> Further, in exploration we have seen that agents go in separate direction and have individual rewards/sense of exploration which usually leads to faster exploration of an unexplored area. Thinking in simple terms, exploration of an house would be faster if different people handle different rooms. Achieving this is hard in CommNet because global rewards don’t exactly tell your individual contributions if you had explored separately. Also in CommNet, we have observed a pattern where agents get together at a point and start exploring together from there which delays the exploration for the enemy/prey. We have updated the paper to reflect this hypothesis and its confirmation using the “attack closest” heuristics baseline in Section 6.2.1.
>
> Note: Macro-actions corresponds to “right click” feature in StarCraft and Dota in which a unit can be called to attack on other unit where units follows the shortest path on map towards the unit to be attacked and once reached starts attacking automatically, this essentially overpowers “attack closest” baseline to easily attack anyone under full-vision without any exploration.
>
> Note 2: You can observe the pattern for CommNet in PP, we just talked about at https://gfycat.com/IllustriousMarvelousKagu . This video has been generated using trained CommNet model on PP-Hard. Red ‘X’ are predators and ‘P’ is the prey to be found. We can observe the pattern where the agents get together to find the prey leading to slack eventually

---

> ### Author Response · Authors · 2018-11-30
> **Request for feedback**
>
> Hi Reviewer1,
> Thanks once again for the great feedback on our work. As suggested, we have made our code available at https://github.com/IC3Net/IC3Net . If you have any updated questions/feedback/thoughts about our paper following our response, we would be happy to address them. Thanks :) .

---

### Official Review · AnonReviewer3 · 2018-11-13
**Interesting work**

**Rating:** 7
**Confidence:** 3

**Review:**

The authors propose a new network architecture for multi-agent reinforcement learning. The new architecture addresses three issues: (1) the applicability of existing algorithms to semi-cooperative or competitive settings; (2) the ability to use local rewards during agent training; (3) the credit assignment problem with global multi-agent rewards. The authors address these issues with a new architecture that is comprised of several LSTM controllers with tied weights that transmit a continuous vector to each other, and that are augment with a gating mechanism that allows them to abstain from communicating.

I think that this paper makes a solid contribution over the existing literature. My main comments are the following:
* I feel like the paper can be strengthened by comparing to additional baselines. The authors compare mainly to Sukhbataar et al., but I think a more detailed comparison to other approaches (e.g. Foerster et al.)
* One of the advantages of this method is that it can be used in non-cooperative settings. I am not familiar with this regime, and I would like a better explanation about why we would train competing agent with the same controller, rather than using a different controller for each team.
* In several experimental results, the proposed method seems to have significantly higher variance than the baselines. I would like to see some discussion about why it is the case.
* Also, in some places (e.g. Table 1), the method is highlighted in bold, even though it doesn’t actually outperform the baseline. Please correct this and only highlight the best method (if several methods are tied, either highlight them one, or don’t highlight any).
* Also, in some cases when the error bars contain the previous best result, I am not sure if we can say that the proposed method is obviously better.

---

> ### Author Response · Authors · 2018-11-26
> **Thank you for your detailed feedback and review**
>
> Dear Reviewer,
>
> We thank you for your insightful and helpful comments. We have responded to your comments, suggestions and concerns inline below. We hope our answers clarify your concerns.
>
> 1. I feel like the paper can be strengthened by comparing to additional baselines. The authors compare mainly to Sukhbataar et al., but I think a more detailed comparison to other approaches (e.g. Foerster et al.).
>
> CommNet and Foerster et. al. are similar approaches based on continuous communication. But different from our framework, Foerster et. al. is based on Q-learning, which makes it difficult to perform a fair comparison since our approach is based on policy gradient. This is due to the fact that there is no straightforward way to compare the sample complexity of Q-learning and policy gradients method because of the replay buffer. For the same reason, we didn’t compare with other approaches in StarCraft multiagent systems because those were defined for different environments than ours making fair comparison even harder.
>
> 2. One of the advantages of this method is that it can be used in non-cooperative settings. I am not familiar with this regime, and I would like a better explanation about why we would train competing agent with the same controller, rather than using a different controller for each team.
>
> Using same controller over different controller leads to shared weights which is reasonable given that all of the agents are performing the same task. Now, instead of training multiple weights, training single weights leads to faster training times and convergence which is beneficial. Further, we would like to point that our framework is independent of weight sharing and can even be used in scenarios where weights are not shared.
>
> 3. In several experimental results, the proposed method seems to have significantly higher variance than the baselines. I would like to see some discussion about why it is the case.
>
> We agree with the reviewer about the significant variance in IC3Net than the baselines. We would like to reiterate our response to R1 on a similar question about variance:
>
> We have performed a lot of experiments on StarCraft and can attribute the  significant variance to stochasticity in the environment. There are a huge number of possible states in which agents can end up due to millions of possible interactions and their results in StarCraft and we believe it is hard to learn each one of them. This stochasticity variance can even be seen in simple heuristics baselines like “attack closest” (Win ratio 76.6 +- 8 calculated over 5 runs)  and is in-fact an indicator of how difficult is it to learn real-world scenarios which also have same amount of stochasticity.
> Albeit, one can ask why don’t we see similar variance in CommNet and others. We believe that this might be due to the fact that adding gating action increases the action-state-space combinations which yields better results while being difficult to learn sometimes. Other point to be noted is that this variance is generally seen when the Win %  (requires learning more states) is above some particular threshold which is close to nothing in baselines. We have added a section 6.2.2 on variance to the paper.
>
>
> 4. Also, in some places (e.g. Table 1), the method is highlighted in bold, even though it doesn’t actually outperform the baseline. Please correct this and only highlight the best method (if several methods are tied, either highlight them one, or don’t highlight any).
>
> We thank reviewer for pointing this out. We have fixed this in the updated version.

---

> ### Author Response · Authors · 2018-11-30
> **Request for feedback**
>
> Hi Reviewer3,
> Thanks once again for feedback on our work. We were wondering if you have any updated thoughts/feedback/questions following our response. We would be happy to address them. We have also made our code available at https://github.com/IC3Net/IC3Net . Thanks :)

---

> > ### Comment · AnonReviewer3 · 2018-12-16
> > **Thank you for your response**
> >
> > I acknowledge that I have add the authors' response. They clarify some questions I had; however overall my score is unchanged.
> >
> > I was especially unconvinced by the response to my Concern #1. I don't understand why a policy search method cannot be compared to a Q-leaning based method. I'm not suggesting that one method has to outperform all others on all criteria (e.g. sample complexity and asymptotic reward). However, a strong paper requires a detailed discussion of the pros and cons of a new method relative to other approaches, ideally supported by data. If existing methods are not applicable, then there should ideally be other baselines, or some kind of ablation analysis that reveals the strengths and limitations of the method. I am generally happy with accepting this paper, but I think the above is necessary to make this a really strong paper.

---

### Author Response · Authors · 2018-11-26
**General Response to Reviewers**

We thank reviewers for their insightful comments and suggestions. We hope our rebuttal clarifies your concerns. We summarize the major updates in this thread:

1. We are working on code release which will be available at https://github.com/IC3Net/IC3Net . We will update the thread when we have published it.
2. We have rephrased and improved coherency in some sections as suggested by the reviewers (Section 4, Tables etc.).
3. We have added new section on analysis of patterns in our results to Appendix 6.2 which includes hypothesis on variance in IC3Net results for StarCraft, explanations of StarCraft Combat results and CommNet results in StarCraft explore task (including a video from run on PP-hard).
4. We have added results on Predator-Prey cooperative scenario to Appendix 6.3 which show that IC3Net can work equivalently or better than CommNet even in case of cooperative scenarios.
5. We have added a new subsection on performance of IRIC and IC to Appendix 6.4.

---

> ### Author Response · Authors · 2018-11-30
> **Code open sourced**
>
> As promised, we have released our code on https://github.com/IC3Net/IC3Net . We have added a simple starter reader which can be used to run most of our experiments.
>
> Along with that, a video showing an episode of Predatory Prey Hard Mixed Version is shown at https://gfycat.com/SimplisticInbornBluetonguelizard
>
> Similarly, a video showing an episode of Traffic Junction Hard version is shown at: https://gfycat.com/MilkyWhimsicalEagle
>
> We hope these links further resolve your concerns/questions. Your feedback has already been very useful in improving the paper. We look forward to any additional comments/questions/feedback you have regarding our paper.

---

### Author Response · Authors · 2018-12-08
**End of discussion period - Thanks for the feedback and review**

As the discussion period is coming to end, we would like to once again thank the reviewers for their feedback which has helped a lot in improving our paper. Let us know about any additional questions/thoughts/feedback. We would be happy to address them in detail.

Best,
Authors.

---

### Meta-Review · Area_Chair1 · 2018-12-19

**Confidence:** 4
**Recommendation:** Accept (Poster)

**Metareview:**

All reviewers agree that the proposed is interesting and innovative. One reviewer argues that  some additional baseline comparisons could be beneficial and the other two suggest inclusion of additional explanations and discussions of the results. The authors’ rebuttal alleviated most of the concerns. All reviewers are very appreciative of the quality of the work overall and recommend probable acceptance. I agree with this score and recommend this work for poster presentation at ICLR.